# Tragedies, Fates, Furies and Fuels: Narratives of Individuals Bereaved by Suicide

**DOI:** 10.3390/ijerph19148715

**Published:** 2022-07-18

**Authors:** Diane Macdonald, Alexandra Nicolopoulos, Kathryn McLachlan, Stephanie Habak, Helen Christensen, Katherine M. Boydell

**Affiliations:** 1Black Dog Institute, University of New South Wales, Randwick, NSW 2031, Australia; a.nicolopoulos@blackdog.org.au (A.N.); k.mclachlan@blackdog.org.au (K.M.); s.habak@blackdog.org.au (S.H.); h.christensen@blackdog.org.au (H.C.); k.boydell@blackdog.org.au (K.M.B.); 2School of Psychiatry, Faculty of Medicine, University of New South Wales, Randwick, NSW 2031, Australia

**Keywords:** bereavement, suicide, grief, qualitative methods, narrative analysis, lived experience

## Abstract

Suicide is the leading cause of death for Australians aged 15 to 44, with fifty to sixty per cent of individuals who die by suicide ‘flying under the radar’, dying in this way without receiving formal mental health care or treatment. This paper explores how people bereaved by suicide interpret and narrate the lead-up to, act and aftermath of a male family member who died by suicide. We used qualitative semi-structured interviews to explore how narratives of suicide were articulated by loved ones bereaved by suicide. Analytic findings were conceptualised through Bamberg’s four layers of cognitive narrative structure–setting, complication, resolution, coda. We derived three complications conveyed by the group as a whole: that the men felt sentenced by fate, charged with fury and were fueled by alcohol. These narratives by individuals bereaved by suicide draw us into the larger picture of meaning-making, the loss of life and finding closure. They also speak to the need for early interventions, as most of these stories are rooted in childhood tragedy that was not sufficiently addressed or supported.

## 1. Introduction


*[E]very good story must have a beginning, a middle and an end, with the end foreshadowed in the beginning.*
-Miranda Cowley Heller [1] (p. 387)

One ending in this paper is foreshadowed from the start. The tragedy of a loved one’s suicide represents a closing chapter for one and the beginning of a new one for another. Listening to the stories of individuals bereaved by suicide can lead to a broader understanding of suicide and the shortcomings of our community care systems in preventing it [2]. This paper explores how people bereaved by suicide interpret and narrate the lead-up to, act and aftermath of a male family member who died by suicide. Suicide is the leading cause of death for Australians aged 15 to 44 [3]. Fifty to sixty per cent of individuals who die from suicide ‘fly under the radar’, dying in this way without receiving formal mental health care or treatment [4]. Most have been in contact with health services for reasons relating to their physical health in the days or months preceding an attempt but do not receive help for their suicidal thoughts, mental health problems or personal circumstances [5]. The participants in our study identified the men as ‘flying under the radar’ before dying by suicide. 

Despite the prevalence of suicide globally, individuals bereaved by suicide remain a group that has been understudied [6]. Even less is known about the bereaved whose loved ones are at risk of suicide who are not in mental health care. Insight from loved ones who experienced the lead-up to and aftermath of a death by suicide can offer recipes for change to our mental health systems [7]. Their stories of suicide can inform decision-making to develop better care models for individuals at risk of suicide. This paper presents a cognitive narrative approach to interpreting 12 narratives of suicide: how participants remember, make sense of and describe a tragic event like suicide [8,9]. In the following sections, we explore, interpret and analyse storytellers’ narratives and story arcs to understand their perspectives on their loved ones’ suicides and what we can learn from their lived experiences. 

### 1.1. Context of Our Study 

The 12 stories of a loved one’s suicide were collected as part of a multi-phase study, Under the Radar (UtR), conducted by Black Dog Institute in 2021. UtR employed a mixed-methods approach to explore the demographic and clinical characteristics, personal circumstances, and preferences for services amongst people at risk of suicide, with the aim of developing a comprehensive pathway to care. At the onset of this larger project, Black Dog conducted a systematic review of the published literature examining predictors of dying by suicide without having received professional help. Key risk factors were identified for being under the radar: male sex, both younger and older age, rural location and the absence of a mental health diagnosis [10]. In the second phase, Black Dog surveyed 415 men and interviewed 37 men experiencing suicidal thoughts and/or behaviours to examine their views toward health services and family members and to understand coping styles and preferences for service provision. These methods, however, do not specifically shed light on those men whose suicidal thoughts and/or behaviours led them to die. This current study represents an approach to understanding these events through the lens of those bereaved by the death of a man who was not in contact with formal mental health services at the time of his death. Our paper focuses on the narratives from the bereaved to understand how the overarching structure of their stories are told to gain insight into their lived experiences of suicide [8]. 

### 1.2. Reflexivity 

Reflexivity involves exploring one’s own situatedness, background and perspectives in relation to the research topic, participants and research processes and practices [11,12]. The authors of this paper are women researchers with lived experience of suicide and expertise in qualitative research, mental health, arts-based methods, knowledge translation, co-design and community health. All interviewers had their own lived experience of suicide. We acknowledge our privileges and gendered subjectivities have influenced our understandings and interpretations of men’s suicidality, including our socially constructed ideas of masculinity and men’s health risks [13]. Qualitative research explores the nuanced, richly textured and deeply contextual lived experience of people as they lived their lives and does not lend itself to a neutral comparison of data [14]. By involving multiple researchers in this study, we hope to contest and strengthen each other’s findings to mitigate our subjectivities in the analysis and report writing [15]. 

### 1.3. Research Questions 

What are the personal experiences and insights of loved ones bereaved by the loss of male friend or family member to suicide who was not in contact with mental health services? 

How do the bereaved describe the lead up to, the act and the aftermath of a loved one’s suicide? 

What meaning is conferred by the bereaved onto a loved one’s suicide? 

## 2. Materials and Methods

This paper investigates how bereaved participants interpret and narrate the lead-up to, act, and aftermath of a loved one who died by suicide. Stories and narratives (used interchangeably here) help us make sense of events [16]. Narratives offer a portal into lived experience of events that incorporate interpretations and subjective meaning onto those events [8]. Events that may be random and chaotic or related and connected are constructed into meaning through the active creation of narrative patterns or order [9]. Narrative analysis aligns people with the stories they tell, offering researchers a rich source of knowledge [17]; and a more in-depth understanding of people and their experiences [18]. Narrators are, therefore, active agents who incorporate a beginning, middle and end, locating their stories in a social context at a specific time and place, adding some evaluative, relational and personal aspects [19]. 

This study used qualitative methodology to collect data through photo elicitation and semi-structured in-depth interviews. Through its exploration of lived experience in its natural environment, qualitative inquiry can contribute to a broader understanding of knowledge [2]. This study explored how narratives of suicide were articulated by loved ones bereaved by suicide. Semi-structured interviews were conducted online with 11 individuals who had lost a loved one to suicide; one interview was conducted through emailed written responses. 

### 2.1. Photo Elicitation 

Prior to the scheduled interview, participants were offered the option of bringing along a photograph that held meaning to them vis-a-vis their experience. All participants chose to do so. At the start of each interview, the participant showed and described a cherished photo of the deceased. Photo elicitation is an arts-based methodology used in qualitative research, whereby photographs encourage conversation about the research topic at hand [20]. It is a method in which photographs produced by the participant are used as a stimulus to guide the interview. Using photographs with personal meaning—whether explicitly taken for this research interview, obtained from an archive of historical photos or a visual representation of an object which holds significant meaning—can evoke deep emotions, memories and ideas [21]. The photo elicitation in our interviews served as both an interview stimulus and an interview guide, allowing the interviewer to obtain insights and rich context which may not otherwise have been obtained. The photos in our study were used as a prompt and were not part of the analysis or data collection. 

### 2.2. Semi-Structured, In-Depth Interviews 

A co-design approach with people who have identified lived experience of suicidal thoughts and/or behaviours was utilised to develop the guideline questions that form the semi-structured, in-depth interview. Co-design between the research team and lived experience advisors contributed to developing suitable and relevant prompts throughout the interview, aligned with the research questions and goals. See Appendix A for interview questions and prompts. 

Questions included: I would like to explore the meaning behind the photos you have taken/provided. Can you please tell me about them?Can you please tell me about the experience/s of your loved one?What kind of support would have been helpful for your loved one?If you could envisage something that would have worked for your loved one in difficult times, what would that look like?In what ways do you feel you have been able to share your loved ones’ experience in this interview?

### 2.3. Data Collection 

Participants were recruited through social media advertisements (Facebook, Twitter, Instagram and LinkedIn) and external partners and organisations. Recruitment took place from August 2021 to October 2021. Participants were required to be 18 years of age or older, living in Australia, comfortable with the interview being conducted in English, and to have experienced a male family member or friend who died by suicide and was *not* in contact with mental health services at the time of his death. 

Interview participants were given an option to receive a call from a clinical psychologist within two business days, if they required additional support, to mitigate risk. None of the participants required additional support. At the cessation of the interviews, all participants were sent a ‘thank you for participating’ email, including the contact information for Lifeline, MensLine and/or Suicide Call Back Service. Participants were offered a reimbursement of $170 as an acknowledgement of the time and effort required to prepare for, attend, and complete the interview and the significance of the value their story is bringing to the research. 

After one of the first interviews, the participant, who works in suicide postvention and is bereaved by suicide, suggested that the interviews be conducted by an interviewer with experience of being bereaved by suicide. They suggested that those who have been bereaved may be more comfortable with, and open to, sharing their experiences with someone who has also experienced bereavement. This suggestion was taken on board by the interview team, and subsequently, nine of the twelve interviews were undertaken by a team member (KM). Author AN and a team researcher completed one interview each. The email (*n* = 1) interview took the form of written responses to the same oral questions and prompts as the participant found it too painful to speak about the experience of their loved one but also wanted to contribute and be a part of the project. Digitally recorded interviews were transcribed through artificial intelligence, checked for accuracy, cleaned and de-identified. 

### 2.4. Sample 

We highlight that 11 of the 12 interviewees were women and acknowledge that the stories of men’s suicides were told through a predominantly female lens. These women were aged 31 years and older and the one male participant was aged 45 (One father joined a mother later in one interview, but no demographics were collected from him.). All (*n* = 12) self-identified as Caucasian and urban with stable living conditions. Participants were de-identified and assigned a letter and a number (i.e., P1–P15), reflecting the fifteen participants who initially agreed to take part, the three who opted out (P5, P8 and P13) and the twelve who completed the interview. The loved ones of the bereaved were sons, husbands, fathers or brothers, see Table 1.

### 2.5. Analysis 

Reflexive thematic analysis offered an organic approach to analysis, as it allowed for a broad and flexible application of the analytic approach [11]. Our team drew upon both semantic and latent theme identification. Semantic themes were descriptive level themes, wherein the content of the data was identified and summarised. The content of the data captured the surface meaning, reflecting what was explicitly said. Latent theme identification at the interpretive level allowed the team to go beyond what was explicitly said, revealing the underlying ideas, assumptions and conceptualisations within the data. 

Our team followed the six phases of reflexive thematic analysis, where we engaged in an iterative process of reflexivity; a process of self-examination; revealing ourselves as individuals and as researchers while understanding how our personal world views may impact the research process [11]. This process of acknowledging our positionality was practised throughout the research. 

Rigour in team analysis was addressed by applying well-established trustworthiness criteria: through prolonged engagement with the subject matter, persistent observation of experiences and perspectives about suicide and researcher triangulation [22,23]. Team discussions in regular analysis meetings included reflexive, recursive engagement with the data and continual acknowledgement of the research teams’ active role in knowledge production and interpretation. 

‘After a thorough coding process, we identified common storylines occurring in the data. Therefore, we analysed the bereaved accounts of suicide through their cognitive structure: the plots, themes, and coherence expressed by participants [8]. We incorporated a bottom-up, inductive construction process, at first, that later reflected a top-down, deductive structure to analyse the relationship between narrative phrases/words and the broader picture of suicide. From 131 codes and subcodes, we focused on shared descriptions of common (a) pain points (triggers, abuse, burdens, trauma, tragedy and exposure to suicide); (b) feelings (helplessness, hopelessness, powerlessness and anger) and (c) coping mechanisms (actions, substance use and exercise) which guided our interpretive conclusions about bereaved stories of suicide. 

## 3. Results

We constructed our analytic findings through Bamberg’s [8] four layers of cognitive narrative structure. We chose this framework as it most closely aligned with the storylines formed by the bereaved and conveyed the substantive part of their narratives. Bamberg’s [8] stories are comprised of: An orientation or settingComplicationResolutionCoda or closure

We re-coded our analysis of shared descriptions to align with these four layers, see Table 2.

In ten out of twelve interviews, we constructed and connected a story arc of tragedy, fates, furies and fuels, see Figure 1. Two participants (P11, P14) told a different story from that of the ten–one of a sudden snap after an argument or event that led to their loved one’s suicide. Both men were in their twenties when they died and a parent supplied the narrative. While rich and valuable, their data have not been included in our findings. The following sections examine each layer of the story arc with narrative examples from the interviews. 

### 3.1. Setting: Childhood Tragedy 

Participants described their loved one as having experienced abuse and/or trauma in ten of the twelve narratives. This constancy across narratives established a setting of abuse or trauma, mainly in childhood (*n* = 9), that impacted their loved one. 

Seven participants (P2, P4, P7, P9, P10, P12, P15) described some form of abuse, either expressly or implicitly: 


*So his parents were terrible every single day every every day. And I’m not, I’m not like I’m exaggerating, his dad would leave a voicemail on his phone like you’re a ******* ****. You’re a ******* hacker stole my ******* business. You done this.*
(P2)


*His father was ten years older than his mother, there was, it was a domestic violence relationship with his parents...and his father would rape his mother really while she was in that kind of state, and then send her back to the institution again. His father was abusive. Uh, my, my mother did say that his father sexually abused him, but I’ve never heard that from my father, and I’ve never heard that from anyone else.*
(P10)


*[He] was bullied at school…He was really chubby. And then as he got older he grew out. You know how they boys do that? There was these kids on the football team and they gave him so much grief, called him nugget and that would upset him.*
(P12)


*[He] had a pet rabbit when he was young and his brother killed it and and and he’d told me about that pet rabbit and told me that a fox killed it and so I think there was some pretty horrific maybe abuse that went on.*
(P15)

Eight participants (P1, P3, P4, P6, P7, P9, P12, P15) set scenes of trauma occurring in their stories of suicide: 


*You know, if the same thing happened now, when you know, if a young child saw their sibling die? There would be so much support for them and they would be monitored and they’d be looked after and all that sort of stuff.*
(P1)


*Childhood trauma I think. He had a very, very different, I guess, childhood, where he was often left alone or without support.*
(P7)


*And then my father and mother died three weeks before he did, and he just didn’t cope well with it.*
(P12)

### 3.2. Complications 

A complication (or complications) is the part of the story where something happens, typically an issue or problem for the main character, that triggers a chain of events [8]. The series of events tells how the characters react to the complication; how rising tension occurs, leading to a high point or major drama and resolution. We derived three complications conveyed by the group as a whole: that the men felt sentenced by fate, charged with fury and were fueled by alcohol. Each complication is explored below. 

#### 3.2.1. Sentenced by Fate 

Five participants (P3, P4, P7, P10 and P15) constructed a narrative arc that indicated the loved one considered themselves to be doomed or diminished by fate: 


*Some of us are just not meant for this world.*
(P3)


*He really was sort of like resigned to, well this is me; you know this is this is me. This is my lot.*
(P4)


*He believed that no one could help him because that’s just the way the world was. This was a fact.*
(P7)

#### 3.2.2. Charged with Fury 

Eight participants (P2, P3, P4, P6, P7, P9, P12, P15) described their loved one as being charged with fury, signaling emotive, instinctual or irrational behaviours: 


*He used to headbutt walls and get into fights because he was so angry and didn’t understand why those thoughts were in his head.*
(P3)


*Stemmed from feelings of powerlessness and then getting so angry.*
(P4)


*He would just be very negative and um, angry. I guess. Um, with… life and the world. He’d express negativity about everything.*
(P7)

#### 3.2.3. Fueled by Alcohol 

Seven interviewees (P1, P2, P3, P6, P9, P12, P15) spoke of problematic substance use (drugs and alcohol) by their loved one: 


*He was, he was happy to drive around and doing all that while I was while he was drunk on drugs and drinking. So, I guess in hindsight like yeah he’s alcohol and drug use really like increased.*
(P2)


*I think it was the alcohol that perhaps allowed him to express that this is the way that he was feeling. And this is [what] you wanted to do to end that pain.*
(P6)


*[H]e was drinking a lot and I didn’t like that and he I’m still finding wine bottles hidden around the house.*
(P15)

### 3.3. Resolution: Suicide

Resolution refers to an action or action orientation, in this case, the suicide of a loved one [8]. 


*[He’s] wrapped some chains around himself and got in the pool or something and she’d gotten home and... she’d kind of found him. He wasn’t. It obviously hadn’t worked. She’d gotten home, and he was in the process of doing it, or had just tried to do it, or whatever… It might have been three or four days before he passed away. He was in intensive care in hospital.*
(P1)


*But he always he always had...looked down on people who were, who were, who were who had suicided because, so in that instance, I was like, oh he’s fine. Then at the weekend that he killed himself we had been arguing and he had said a few things like I’m gonna go away. I’m gonna go away. But I didn’t really think of it as that…I was telling him to leave me like he was being so intense and he got a knife, but I like I didn’t realize the, I didn’t realize. But he’d never actually like vocally voiced that he had suicidal intent or suicidal ideations.*
(P2)


*He didn’t have his seatbelt on, and he lined his car up with the pole and drove into it at 180 kms an hour. In the video he explained that he had been having thoughts about wanting to end it for years. He said that the last few months, the thoughts had taken over his life and none of his distractions were working anymore.*
(P3)

### 3.4. Coda: Making Sense out of Chaos and Tragedy

A coda underlines how the character has changed and what they learnt as a result of the experience they went through [8]. The coda takes the listener/reader away from the act of suicide and back to the present.


*Yeah, I think it was going to be hard for him as a 48-year-old [42 years later] decided now just access this, um, these services, having never done it before, I think you know there was obviously, you know, all of those years of resistance and, you know, there’s just so much to unpack for him. Yeah, I just feel like, you know, if it had been a process that was, you know, gradual from when he was young all through his life you know it would have been easier.*
(P1)


*You realise that it was their decision and they did it. And if they didn’t do it then, they would have done it some other time. That’s my thought.*
(P7)


*I think I could have helped him more. I may not have stopped it, but I’m sure that I was dealing with it so wrong. The best you know, and I think that’s the only way I can get through this, as I’ve just had to find peace with it and and light and not going to that dark place and just go. He’s found peace and this was obviously [his] journey and for whatever reason this is also my journey.*
(P15)

## 4. Discussion 

This paper aims to make meaning from a group of narratives that describe the lead-up to, act, and aftermath of suicide. The participants in our study, mainly women, were family members of the decedent who identified that their loved one died without receiving formal mental health care or treatment. In this context, we explore their unique viewpoint to spark life-saving changes to our mental health and community care systems [2,7]. Their stories of suicide can inform better decision-making to help individuals at risk of suicide. Insight into their collective story of suicide can facilitate the development of services and pathways to better care.

Incorporating photo elicitation at the start of each interview acted as a stimulus to guide conversation [21]. Qualitative inquiry develops a broader understanding of social events in their natural context [2]. Through a rigorous analytical process, we were able to determine a common cognitive pattern and story arc through ten of the twelve interviews [8,9]. The participants in our study established a setting that involved tragedy, including abuse and trauma, mostly in childhood. These tragic encounter(s) had long-lasting effects that were described as diminishing or enraging the decedent. Childhood trauma, including physical, emotional, and sexual abuse and physical neglect, is noted to be a modifiable risk factor for suicide [24]. Problematic substance use added a further complication, echoing research findings of the strong relationship between suicide and substance abuse [25]. The resolution, a final act taken by each of the men, was foreshadowed at the start. Developing this grim story arc leads to important implications for men who ‘fly under the radar’.

Tragedies and tragic events will continue to occur. Watching the nightly news, alone, provides evidence of daily death, disaster and destruction. How we as individuals and communities engage, react and learn from these events can be key. The men in our study were not taking part in any formal mental health care, reflecting the experiences of over half of all people who die by suicide in Australia [4]. Trauma and abuse mostly occurred in childhood, with little or no support at the time, suggesting that early intervention through community response will offer an avenue of help. Resilience training, equipping men with skills and knowledge to recognise feelings and learn to develop appropriate responses, can also help and support those at risk. However, the ‘under the radar’ men are difficult to reach by the very nature of their experiences. Connecting with men in innovative and non-stigmatising ways, through a variety of non-health settings such as workplaces, schools, social welfare agencies, cultural organisations and social media is needed [10]. As participant P1 described earlier in this paper, there is currently a greater community response to the needs of young people impacted by tragedy, abuse and trauma than ever before.

So far in this discussion we have focused on the lead-up to and act of suicide, through the eye of the bereaved. It is equally valuable to explore the codas, the aftermath, and how the bereaved described what they learnt as a result [8]. Participant P15 voiced a need for better skills and knowledge of managing a loved one’s mental illness (*I was dealing with it so wrong*). P1 indicated that early intervention may have helped her loved one (*all of those years of resistance* [to seeking help]). P7 was resigned to the inevitability of her loved one’s suicide (*if they didn’t do it then, they would have done it some other time*). The first two indicate mental health/suicide education and communication can play a positive role in the future. Further investigation into their insights is warranted.

## 5. Limitations

We acknowledge that the stories told by family members bereaved by suicide may not reflect the reality of the men’s experiences or actual events. We also note that the interviewees were largely white urban women, like the authors of this paper. A lack of diversity in participants suggests that further research could focus on a more diverse group of the bereaved. Some work has begun to focus on the experience of bereavement by suicide in ethnic minority groups as in Rivart et al.’s [26] article in this special issue. Our findings were also limited by the gendered lens that the women participants and women researchers overlaid on stories of men’s suicide that may help sustain damaging masculinity tropes [13]. Input from men researchers and more men participants would have been a welcomed perspective. In addition, a differing story arc, as told by two of the participants (one of a sudden snap after an argument or event that led to their suicide), is worth exploring to broaden our understanding of suicide and pathways to suicide prevention.

## 6. Conclusions

A narrative analysis, to be merited on its worth, must be grounded in rigour and a thorough review of the data [18]. Through prolonged engagement with the narratives, persistent observation of experiences and perspectives about suicide, and researcher triangulation, we constructed an overarching storyline from ten stories of suicide. Consolidating ten stories into one cohesive story arc can help us see through the complexity of components that play into a decision as grave as suicide and make sense of events [16]. Expanding our knowledge of their experiences will drive change in our community care systems [2].

These narratives by the bereaved draw us into the larger picture of meaning-making, the loss of life and finding closure. They also speak to the need for early interventions, as most of these stories are rooted in childhood tragedy that was not addressed or supported. Our community care systems have improved over the years, but there is a need to develop greater resources to equip our young people with the skills and knowledge to build resilience. Decision-makers, practitioners and researchers can use this collective story of tragedies, fates, furies and fuels to broaden their understanding of individuals at risk of suicide and develop better models of care.

## Figures and Tables

**Figure 1 ijerph-19-08715-f001:**
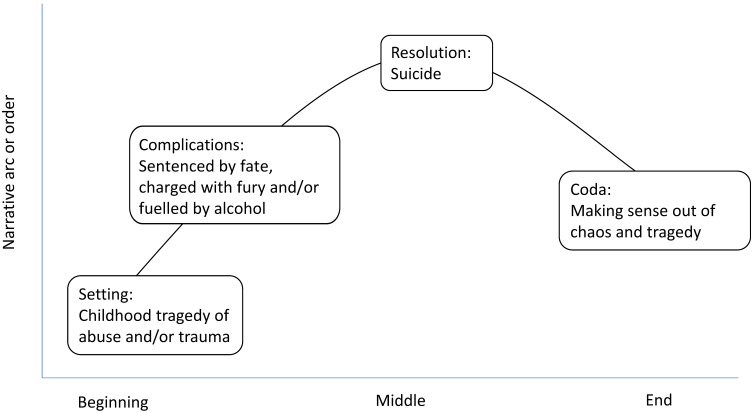
Story arc of suicide.

**Table 1 ijerph-19-08715-t001:** Participant relationship to the decedent.

Participant Reference	Participant’s Age	Decedent’s Details
P1	45	brother died in 2019, age not stated
P2	31	partner died in 2014 when he was 25 years old
P3	38	relationship unclear, died around 2020
P4	74	husband died in 2013, age not stated
P6	47	father died in 1988, age not stated
P7	63	husband died in 2004 when he was 44 years old
P9	52	son died in 2018, age not stated
P10	48	father died in 2011, age not stated
P11	44	son died in 2018, age early to mid-20 s
P12	51	son died in 2017 when he was 18 years old
P14	58	son died in 2018 when he was 26 years old
P15	44	husband died in 2021 when he was 50 years old

**Table 2 ijerph-19-08715-t002:** Narrative alignment.

Four Layers of Cognitive Narrative Structure	Initial Thematic Analysis Heading	Narrative Analysis Heading
Setting	Pain points	Tragedies
Complications	Feelings and coping mechanisms	Fates, furies and fuels
Resolution	Suicidal thoughts and behaviours	Suicide
Coda	Looking back	Making sense out of chaos and tragedy

## Data Availability

The data is not publicly available as per our ethics approval.

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
