# Peer review of "Tragedies, Fates, Furies and Fuels: Narratives of Individuals Bereaved by Suicide"

_ijerph, 2022, doi:10.3390/ijerph19148715_

Round 1
Reviewer 1 Report
The authors employ what I recognize as the theory of narratology to organize the remembrances of the bereaved. It's a good approach, but the methodology gets in the way of a good story. Longer excerpts and a little less methodological anxiety would be helpful. In the end it's the stories, not the method, that we want to hear. I recommendation publication. To repeat: the authors should worry less about method, more about telling a story.
Author Response
Thank you for your succinct and on-the-nose comments. We have tightened the methods section and developed some of the participant's stories, without making the paper overly long. We've also addressed the other reviewers' comments, including greater emphasis on the gendered perspective of masculinities and our mostly women participants/authors.
Reviewer 2 Report
The authors present qualitative data on suicide survivors= family members, in a mostly female, caucasian sample in Australia and provide a model of how they experience and describe the suicide. I have myself no experience with the specific qualitative approach used, but it looks reasonable, though method and sample size limit generalisation of results, so this might be seen as a pilot for further research. Information by family members, such as on childhood trauma of the suicide victims is hearsay and might just confirm the narrative and coping or self-blame strategies of the family members, and might not reflect actual events contributing to suicide, this should be made clear. Please describe in more detail if any self-blaming process was observed or excluded, and if you have any data on reactive trauma reactions (indirect trauma). This is a form of psychological autopsy study, and there are other studies on the impact on family members and on the victims with this method, that should be summarised in the beginning, and also on child hood trauma and sucide risk, that again should be summarised in the discussion.
Ethics votum was taken for the study.
I recommend clarifying the above points.
There might be small typing errors (see for example "The men in our study 376
were not taking part in any formal mental health," do you mean did not receive any mental health care
Author Response
Thank you for your review and comments, they were really helpful and combined with the two other reviewers' comments, have strengthened this paper. We appreciate your help.
- We have further clarified that the narratives might not reflect actual events.
- We strengthened our argument on childhood trauma and suicide risk, as well as substance abuse and suicide.
- Great spot on the typo, we have corrected it and a few other minor typos.
- We did not introduce the topic of the self-blaming process as we felt it would add a complication to the narrative that was out of scope. However, we have taken this concept on board as we begin our comparative analysis of the interviews with men and the bereaved. We think it will be a beneficial analytical lens, we thank you for that.
Reviewer 3 Report
The study is well conducted and presented. I have just two suggestions:
1. It would be good to know how long ago the suicides happened, because the experience and the reflections about it change on time. It would also good to have a table describing each respondant relationship to the men they lost, because the reader can interpret the codes better - I was thinking: did she lost her son or husband? I think it makes a difference to the reader.
2. It is not a coincidence that almost all the respondants are women. A gender perspective on masculinities could improve the manuscript, and also help to analyse the interviews. The themes are very related to what masculinity studies show. and discussion could be improved using this literature.
Author Response
Thank you for your review and comments, they were very helpful and improved our article. 1) Your idea of a more detailed table of participants' relationships to the decedent and date of death has been added. 2) We also highlighted the gendered perspective of masculinities in three different ways, without detracting from the storyline.